# The Spatiotemporal Characteristics of 0–24-Goal Polo

**DOI:** 10.3390/ani9070446

**Published:** 2019-07-16

**Authors:** Russ Best, Regan Standing

**Affiliations:** 1Centre for Sport Science and Human Performance, Wintec, Hamilton 3288, New Zealand; 2School of Health and Social Care, Teesside University, Middlesbrough TS1 3BX, UK

**Keywords:** Polo, GPS, Pony welfare, Horse, Training, Equine, Equestrian

## Abstract

**Simple Summary:**

Polo is an equestrian sport that requires two teams of four players to score goals at opposing ends of a 150 m × 275 m pitch. Each player is rated on a handicap system (−2 to +10) that quantifies their abilities and permits their inclusion in different levels of Polo play; the cumulative handicap of the four players sets the level of play. Using GPS technology, we investigated how levels of Polo differ regarding distance covered, speeds achieved and high-intensity activities performed. As cumulative Polo handicap increased, so too did the distances and average speeds attained, decelerations performed and impacts encountered during each period of play. These findings suggest that as each player improves and increases their handicap, they will need to ensure the ponies they play have sufficient aerobic, anaerobic and speed capacities to perform effectively at that level. This information provides valuable insights to Polo players, grooms and equine vets, as to how they can best prepare their ponies for game-day and how they may be able to maintain pony longevity in the sport.

**Abstract:**

Global positioning systems (GPS) have recently been shown to reliably quantify the spatiotemporal characteristics of Polo, with the physiological demands of Polo play at low- and high-goal levels also investigated. This study aims to describe the spatiotemporal demands of Polo across 0–24 goal levels. A player-worn GPS unit was used to quantify distance, speed and high-intensity activities performed. Data were divided into chukkas and five equine-based speed zones, grouped per cumulative player handicap and assessed using standardized mean differences. Average distance and speed per chukka increased in accordance with cumulative player handicap, with the magnitude of differences being trivial–large and trivial–very large, respectively. Differences between time spent in high-intensity speed zones (zones 4 and 5) show a linear increase in magnitude, when comparing 0 goal Polo to all other levels of play (Small–Very Large; 6–24 goals, respectively). High-intensity activities predominantly shared this trend, displaying trivial–large differences between levels. These findings highlight increased cardiovascular, anaerobic and speed based physiological demands on Polo ponies as playing level increases. Strategies such as high-intensity interval training, maximal speed work and aerobic conditioning may be warranted to facilitate this development and improve pony welfare and performance.

## 1. Introduction

The use of global positioning systems (GPS) in sport and animal research is increasingly prevalent and can provide valuable data pertaining to activity type, distance covered, speeds attained and location [1,2,3,4]. Despite reported widespread use in equine settings [4,5,6], the use of GPS to provide tactical or training value in equestrian sport appears limited or underreported. This may be due to a perceived inability to interpret the data obtained [5,6,7], hence most published GPS use in equestrian settings consists of methodological reports, typically pertaining to reliability [7,8,9,10,11].

In order to advance the application of GPS data in equestrian sports, consistent GPS use in training and competitive scenarios is to be encouraged [6,12]. A greater understanding of the external workloads (speed, distance, accelerations, and decelerations) placed upon Polo ponies would not only inform training and competition management but would also be of benefit to ponies returning from injury [12] or transitioning from one equestrian discipline to another, as individualization of training volume and intensity can be easily assessed and prescribed.

Polo presents an ideal model to apply GPS, as Polo ponies are required to perform high-intensity movements and tolerate impacts in a manner that is unique to Polo, and players are required by Polo regulations to interact with a relatively large number of ponies per game in comparison to other equestrian pursuits [13]. Polo is played on the largest pitch in professional sport (275 m × 145 m); this permits the attainment of high speeds and distances covered by Polo ponies. Furthermore, Polo is seeing an increase in ponies transitioning from racing [14]; this may increase game speed or encourage the tactical use of fast ponies, as well as promoting horse longevity by providing a viable life after racing [14]. Polo players are assigned a handicap (−2 to 10 goals), which provides a quantitative measure of players’ ability based on horsemanship, playing skill (individual and team) and the quality of Polo ponies used [13]. The level of Polo play is depicted by the cumulative handicap of all four players on a team (i.e., 10 goal) and can be made up of various combinations of players and skill levels.

This study aims to assess the spatiotemporal demands of Polo, across a range of handicap levels, to accurately describe the performance requirements placed upon Polo ponies, with a view to informing training practices and identifying points of distinction between levels of play. It is hypothesized that as cumulative player handicap (i.e., level of play) increases, average speed and distance covered per chukka (period of play) will also increase.

## 2. Materials and Methods

### 2.1. Sample Population

All data were gathered during the 2018–2019 New Zealand Polo Season, on the north island of New Zealand. Data were obtained from a total of 338 chukkas of outdoor field Polo. All players had a current New Zealand Polo Association handicap (range −2 to +7 goals). The cumulative handicap for each team (four players) was used to define the level of play (goals) for the tournament (e.g., 0 + 5 + 4 + 7 = 16 goals). All games were contested under Hurlingham Polo Association rules [13] and were played over four chukkas (0, 6 and 10-goal, n = 40, 40 and 80, respectively), with the exception of 16 (n = 154) and 24-goal (n = 24) games, which were contested over six chukkas. Ethical approval for this investigation was provided by Waikato Institute of Technology’s (Wintec) ethics committee (Approval code: WTFE2601102018).

### 2.2. GPS Data Collection

The present investigation utilized VX Sport 350 GPS units (VX Sport, Wellington, New Zealand), sampling at 10 Hz, with a speed range of 0–60 km/h, in equestrian mode. The speed range permits for derivation of speed zones (see 2.3 Data Processing and Analysis) but does not set an absolute upper limit upon data captured. These devices have previously been reported as reliable independent of unit position (CV < 10% and ICC > 0.70 [15]), for use in Polo [7].

GPS units were turned on 30 minutes prior to the start of each game to allow sufficient time for satellites to be located and a secure connection to multiple satellites established. As players use multiple ponies per game, possibly per chukka, with limited time between chukkas, it is neither feasible nor representative of typical Polo play to mount a GPS unit per horse or record data per horse; hence the use of a player-worn unit. Each player was fitted with one GPS unit in a pouch on the player’s belt; this position has previously been shown to produce reliable results of speed and distance in Polo [7], with the same unit assigned to the same player for each data collection to further enhance reliability. The belt pouch was secured with insulation tape to minimize potential oscillation of the unit during data collection. Notational analysis was performed by the researchers during the game to describe start and end time of each chukka, which was used to ‘trim’ data subsequently. Upon game completion, units were collected by researchers and turned off, ending the data collection session.

### 2.3. Data Processing and Analysis

Data were extracted using specialist software (VX Sport, Wellington, New Zealand) and were trimmed to remove the initial satellite lock period. The game period was divided into chukkas as per notational analyses that accompanied each game. Speed zones were assigned a priori based upon an estimated maximum speed of 60 km/h, which is within the tolerable limits of the manufacturer’s equestrian mode. Using in-built software thresholds, the following speed zones were constructed: Zone 1: 0–19.2 km/h; Zone 2: 19.2–23.4 km/h; Zone 3: 23.4–28.2 km/h; Zone 4: 28.2–47.4 km/h; and Zone 5: 47.4–60 km/h.

Distance covered (m) and time (min:sec) in each speed zone per chukka were selected as primary dependent variables, with the number of sprints (a positive or negative acceleration > 3m/s/s), impacts, and acceleration and deceleration counts, collectively termed high-intensity activities, provided as secondary dependent variables that further describe the load placed upon Polo ponies. Data were presented per chukka to allow comparison between levels of play.

Data were exported to Microsoft Excel and variables were analyzed using a customized spreadsheet to calculate standardized mean differences (Hedge’s *g*) ± 90% confidence intervals (C.I.), between handicap levels (0, 6, 10, 16 and 24 goals). Standardized mean differences were described using the following magnitudes: trivial 0–0.2, small 0.2–0.6, moderate 0.6–1.2, large 1.2–2.0, and very large >2.0 [16]. An effect was deemed meaningful if the accompanying C.I. did not overlap zero. Descriptive data were reported as medians, unless otherwise stated, due to the data being non-parametric. The data for this publication are freely available online [17].

## 3. Results

The median chukka duration from the sample (n = 338) was 11:09 ± 0:10 (min:sec), with absolute minimum and maximum values of 6:33 and 19:27, respectively.

### 3.1. Distance Characteristics

Distance characteristics for each level of play are shown in Figure 1; a predominant increase in median distance covered per chukka was seen as cumulative player handicap increased. Large increases in mean distance per chukka were observed when 24 goal Polo was compared to all other levels of play, with average 10 goal chukka distance showing a small increase in comparison to that covered per chukka at 0 and 6 goal levels. All other comparisons either showed trivial differences in average distance covered per chukka or had C.I. that overlapped zero.

Distance covered in each speed zone per chukka at each level of play is shown in Table 1, with all effect sizes, C.I. and descriptors for all comparisons found in Appendix A. As the level of play increases, there is a trend towards an increase in distance covered in higher intensity speed zones. This is most apparent in speed zones 4 and 5, as 24 goal Polo displays a very large increase in distance covered in speed zones 4 and 5 compared to 0 goal play. This increased high-speed distance demand decreases in magnitude when 24 goal play is compared to 6, 10 (large) and 16 goals (moderate). Differences in lower speed zone (zones 1–3) values are predominantly small to moderate across all levels of play; however, large differences between distance covered are seen when 16 and 24 goal play are compared for speed zones 2 and 3. These findings support the general distance characteristics outlined above (Figure 1), suggesting that not only does average chukka distance tend to increase with level of play but the speed at which this distance is covered also increases proportionally.

### 3.2. Speed Characteristics

Average speed per chukka increases in accordance with increasing cumulative player handicap (Figure 2), with the magnitude of differences observed between levels of play also increasing. Large differences in average speed per chukka are seen between 0 and 24 goal play, with average speed between 0 and 10, 0 and 16, and 6 and 24 goals differing moderately. All other comparisons present small differences in average speed per chukka, except for 0 and 6 goal play, which only differ from each other trivially.

Time spent in each speed zone per chukka at each level of play is shown in Table 2, with all effect sizes, C.I. and descriptors found in Appendix A. Broadly speaking, differences between cumulative player handicaps increase in number and magnitude as cumulative player handicap and speed zone number increase. Differences in time spent in Zone 1 are predominantly trivial or have C.I. overlapping zero; however, small reductions in Zone 1 time are seen when 10 goal play is compared to 0, 16 and 24 goal play. In Zone 2, 0 goal play only differs trivially to that of 10 and 24 goals, with 10 and 24 goals also differing trivially to each other. All other Zone 2 comparisons differ by a small to moderate extent, bar large differences between 10 (3:25 ± 0:07) and 16 goal (2:11 ± 0:09) levels. There is a large difference in time spent in Zone 3 between 16 (1:35 ± 0:05) and 24 goals (2:34 ± 0:13), and these levels differ moderately in comparison to 0 and 10 goal play. Six goal Polo shows small and moderate reductions in time spent in speed zone 3, when compared to 10 and 24 goal play, respectively, but ponies were subject to a small increase in speed zone 3 time compared to 16 goal Polo.

Differences between time spent in speed zones 4 and 5 show a linear increase in magnitude, when comparing 0 goal Polo to all other levels of play (small–very large; 6–24 goals, respectively), with a similar trend seen when 6 and 10 goal play were compared to 16 and 24 goal levels (small–large effects); confidence intervals for 6 and 10 goal play overlap zero in speed zone 4, and they differ trivially to one another in time spent in speed zone 5. Confidence intervals also overlap zero when time in speed zone 4 is compared between 16 and 24 goal play, yet moderate differences are also seen when comparing time spent in speed zone 5 between these levels. Collectively, these findings emphasize the findings outlined in Figure 1, showing that differences between levels of play typically increase in magnitude, with increased average playing velocity.

### 3.3. High-Intensity Activities

All effect sizes, confidence intervals and descriptors for high-intensity activities can be found in Appendix A. Table 3 shows a tendency for values of all high-intensity activities to increase as level of play increases. There is also apparent ‘stability’ of values when 0 goal play is compared to 6 and 10 goal levels, with all comparisons showing trivial differences, or confidence intervals that overlap zero. The only exception occurs between 0 and 10 goal levels, where a pony would perform a small increase in decelerations. However, when 0, 6 and 10 goal values are compared to 16 and 24 goal play, small to moderate differences in sprint counts are observed. This increases to a large difference in sprint count when 6 and 24 goal levels are compared, with a small difference in sprint values also seen between 16 and 24 goal values.

Differences in accelerations only occur in 50% of comparisons; 16 goal play requires moderately more accelerations than 0, 6, 10 and 24 goal Polo, with 24 goal Polo only demonstrating a small increase in acceleration count compared to 10 goal play, whereas small to moderate differences in decelerations are seen between all level comparisons, except for 0 and 6 goal levels (48.8 ± 3.3 and 48.7 ± 3.7, respectively; trivial), when 16 and 24 goals are compared (60.5 ± 2.0 and 65.4 ± 5.3, respectively; C.I. overlaps zero). Moderately fewer impacts were sustained in 0 and 10 goal play compared to the 24 goal level (1.2 ± 0.3); this difference decreased in accordance with handicap, and when 0 and 10 are compared to 16 goal (1.2 ± 0.2) play the difference is small. Confidence limits overlap zero between all levels of play and 6 goals, likewise for 0 and 10 goal impact counts.

## 4. Discussion

The aim of this research was to assess the spatiotemporal demands of Polo and to accurately describe and compare the performance requirements placed upon Polo ponies across varying levels of Polo play. It was hypothesized that as cumulative player handicap increased, average speed attained and distance covered per chukka would also increase. The findings of this investigation support this initial hypothesis, with overall trends displaying a rise in distance and speed metrics as level of play increased. Further to this, speed zones 4 and 5 showed a linear increase in magnitude when compared across level of play; a trend also shared by decelerations and impacts. These findings provide valuable insight into the horse management and tactical demands of Polo, as they afford a greater understanding of potential horse welfare considerations and may also mitigate potential injuries to ponies or Polo players.

The use of the cumulative team handicap to categorize Polo encourages creativity and variety in approaches to best satisfy this constraint, whilst maximizing a team’s effectiveness. For example, a 0-goal team may be made up of three players with a -2 handicap, and one 6 goal player; or, equally, it may comprise two 1 goal players, a 0-goal player and one -2 goal player. As cumulative player handicap increases to ≥10 goals, it prompts the inclusion of higher handicapped individuals in order to be competitive. Based on the HPA handicap guidelines [13], a higher player handicap suggests increases in level of ball control and riding ability and the inclusion of more capable ponies across a player’s string of ponies. These factors facilitate the flow of the game, permitting a faster, more expansive style of Polo, as evidenced by higher average speeds (Figure 2) and a greater proportion of distance and time spent at higher velocities (Table 1 and Table 2, respectively) per chukka. Increased handicap will likely also have a strategic influence on gameplay and as such may increase the number of high-intensity activities performed per chukka (Table 3). Collectively, the combination of distance covered at high velocities and increased high-intensity activity counts suggest that as cumulative player handicap improves, there is a concomitant physiological cost upon the players’ ponies. Previous quantification of the cardiovascular demands of low-goal Polo (≤6 goals) has reported that Polo ponies are subject to moderate to high cardiovascular stress [18], with 56 ± 8% of playing time spent at heart rates ≥75% heart rate maximum. This high cardiovascular demand has been corroborated by hematological measures in high-goal Polo ponies, who demonstrated acutely high markers associated with anaerobic metabolism, post-game [19,20].

Gondin et al., [21] concluded that positional attributes may elicit varying energy system contributions in Polo ponies, as defenders displayed elevated blood lactate concentrations and glycolysis markers post-game, indicative of a greater anaerobic contribution during game play. This increased anaerobic contribution may be explained by an increase in high-intensity activities as handicap increases as per this investigation; however, we have previously shown that defensive players tend to be more highly handicapped, and have a greater shot success rate [22], supporting the notion that high-goal players require a string that can meet the tactical and physiological demands of high-goal Polo. From a training perspective, this suggests that as players improve their handicap and play in higher goal Polo matches, there needs to be accompanying improvements in pony fitness and anaerobic capacity. However, there is a documented tendency towards aerobic development in Polo training programs [23], which may alter muscle fiber types to become more oxidative in nature, even within the competition phase of the Polo season [23]. Based upon the somewhat linear relationship between cumulative player handicap, high-intensity demands (Table 3) and time spent in speed zones 4 and 5 (Table 2), we recommend the incorporation of high-intensity interval training, a strategy that has been shown to be effective in thoroughbred race ponies [24], in Polo training programs, although aerobic training should not be neglected as chukka lengths in the present study ranged from 6:33 to 19:27 (min:sec).

By understanding the requirements of the level of Polo being played and the physical capabilities of a player’s string, pony management strategies can be further individualized to maximize the effectiveness of each pony and ultimately improve their contribution to the team’s performance whilst ensuring pony and player safety [25,26]. Practice chukkas may be an effective way of achieving this [7,20,21], and may be more protective than longitudinal high-intensity interval training. Whilst high-intensity interval training may develop anaerobic characteristics, it has been shown to induce premature aging of superficial digital flexor tendon [27]. Alternatively, opting for pony management strategies such as opting to ‘half-chukka’ or ‘cycling through’ one’s string may be appropriate at 16 and 24 goal levels, and support attainment of high speeds and distances as per the tactical demands of the level of play, without compromising athletic pony longevity.

Speed zone (Table 1 and Table 2) and high-intensity activity data (Table 3) were analyzed to provide a more thorough breakdown of the differences observed between levels of play. As the level of play increased, the time spent, and distance covered, in speed zones 4 and 5 increased also. This suggests that higher velocity play, comprised of more frequent decelerations and impacts, is a requisite proportional to cumulative player handicap; at the individual level, this may be a manifestation of improvements in riding and technical abilities and repeated positive interactions with one’s string [28,29]. This is an important finding from a horse welfare perspective too, as high-intensity efforts are common causes of musculoskeletal injuries and tendon injuries and are the most commonly reported injuries in Polo ponies [26]. Whilst up to 91% of Polo players actively check ponies’ tendons prior to exercise [26] and bandaging tendons is compulsory to play Polo under Rule 4c of the HPA rules [13], without appropriate training and conditioning increases in pony workload caused by exposure to high-intensity activities and velocities, ponies may be put at an increased risk of injury. Decelerations likely present the greatest risk of injury due to eccentric loading through multiple joints [29], and potential torques generated if these decelerations are accompanied by turns [30,31]. Impacts may also increase the energetic cost of playing Polo on ponies, but through accompanying notational analysis we feel that despite a linear relationship with cumulative player handicap, the present values may underreport impact occurrence. This may be due to the technical nuance of a ride-off (impact), with a more frequent technique being a sustained application of pressure when contesting the ‘line’, as opposed to a collision-based contact. It is understood that these movements and thus injury risks are an inherent part of Polo. The longitudinal use of appropriate monitoring and performance analysis by GPS as outlined within this paper may be best used to complement established risk management strategies outlined above to increase the health, longevity and playing performance of Polo ponies. The incorporation of notational analysis to describe player pony interactions and assess tactical outcomes may also add value to future Polo research [22].

A possible limitation of this study is the use of a player worn GPS unit, which indirectly but reliably measures the characteristics outlined in this paper [7]. However, this is considered most feasible for Polo as a player potentially undertakes many pony–player interactions per game [7]. The use of a player-worn GPS unit may also permit an investigation into the unique movement signature brought about by individual pony–player interactions, allowing for a thorough kinematic evaluation of riding technique and resultant pony gait. This paper has identified trends and values at a team level; however, future research may seek to investigate how these metrics vary at an individual level to identify the strengths and weaknesses within a player’s string, and how best to train or manage these ponies. Further work is also required to understand whether player position interacts with measures of equine Polo performance in a causative manner.

## 5. Conclusions

The aim of this research was to assess the spatiotemporal demands of Polo and to accurately describe and compare the performance requirements placed upon Polo ponies across varying levels of Polo play. Key findings of this investigation were that as cumulative player handicap increased, so too did distance covered per chukka, with a greater proportion of time spent at higher velocities and a greater number of high-intensity activities also performed. As the level of play increases, the increased average speed and distance covered require ponies to possess the cardiovascular and anaerobic performance/fitness to match the physiological demand of the level of Polo they are playing. Strategies to facilitate this development may include the incorporation of high-intensity interval training, maximal speed work, and aerobic conditioning. GPS presents a tool that can effectively quantify the spatiotemporal demands of Polo, and is capable of detecting changes in activities that are indicative of the level of Polo played.

## Figures and Tables

**Figure 1 animals-09-00446-f001:**
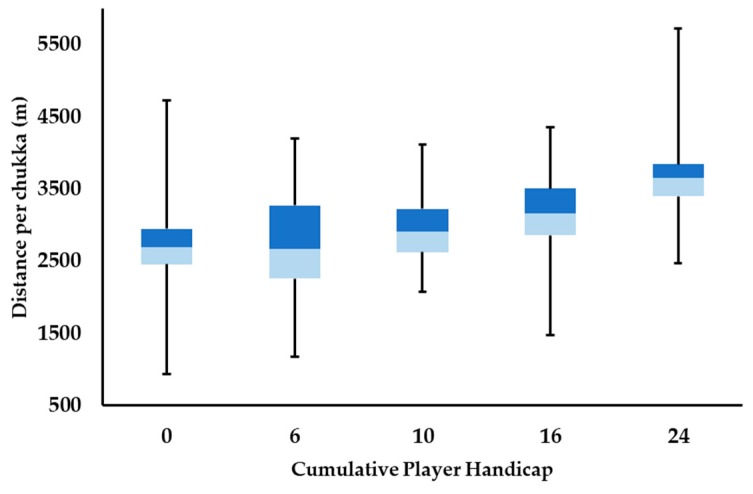
Box-plot of the median distance (m) per chukka at each level of play. Lower and upper box boundaries 25th and 75th percentiles, respectively, line inside box median, lower and upper error lines minimum and maximum, respectively.

**Figure 2 animals-09-00446-f002:**
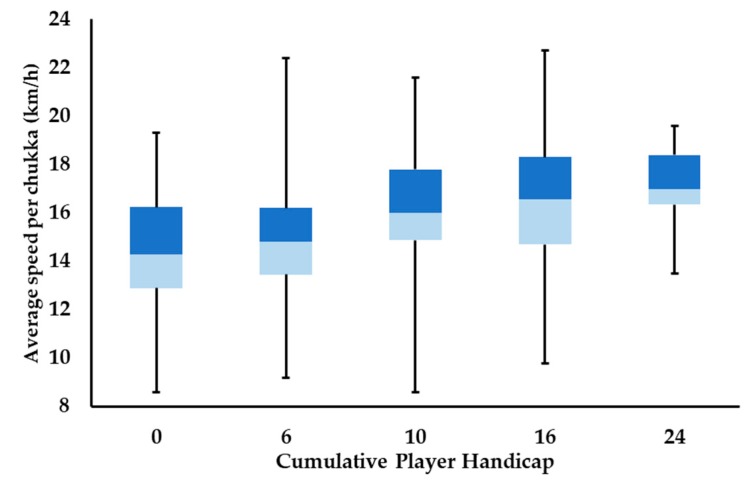
Box-plot of the median average speed (km/h) per chukka at each level of play. Lower and upper box boundaries 25th and 75th percentiles, respectively, line inside box median, lower and upper error lines minimum and maximum, respectively.

**Table 1 animals-09-00446-t001:** Distance (m) covered in each speed zone, per chukka at each level of play. Data are presented as means ± 90% confidence intervals.

Level of Play	Speed Zone 1	Speed Zone 2	Speed Zone 3	Speed Zone 4	Speed Zone 5
0 goal	377.2 ± 27.5	1036.9 ± 72.8	981.2 ± 114.9	287.7 ± 56.6	15.1 ± 8.4
6 goal	410.9 ± 35.2	927.7 ± 55.5	914.9 ± 77.2	397.0 ± 62.1	41.4 ± 17.1
10 goal	381.4 ± 19.5	1044.6 ± 36.5	1003.3 ± 43.5	461.6 ± 43.0	46.4 ± 11.1
16 goal	604.9 ± 34.0	690.7 ± 45.0	744.9 ± 49.2	717.8 ± 43.1	88.6 ± 12.9
24 goal	460.3 ± 34.4	1101.5 ± 92.2	1251.8 ± 108.4	796.4 ± 94.3	150.8 ± 32.6

**Table 2 animals-09-00446-t002:** Time (minutes: seconds) spent in each speed zone, per chukka at each level of play. Data are presented as means ± 90% confidence intervals.

Level of Play	Speed Zone 1	Speed Zone 2	Speed Zone 3	Speed Zone 4	Speed Zone 5
0 goal	5:28 ± 0:27	3:23 ± 0:14	2:02 ± 0:14	0:25 ± 0:05	0:01 ± 0:00
6 goal	5:22 ± 0:27	3:03 ± 0:10	1:52 ± 0:09	0:35 ± 0:05	0:02 ± 0:01
10 goal	4:51 ± 0:17	3:25 ± 0:07	2:04 ± 0:05	0:41 ± 0:03	0:03 ± 0:00
16 goal	5:37 ± 0:14	2:11 ± 0:09	1:35 ± 0:05	1:09 ± 0:04	0:06 ± 0:00
24 goal	5:44 ± 0:22	3:33 ± 0:17	2:34 ± 0:13	1:10 ± 0:08	0:10 ± 0:02

**Table 3 animals-09-00446-t003:** High-intensity activities per chukka at each level of play. Data are presented as means ± 90% confidence intervals.

	0 Goal	6 Goal	10 Goal	16 Goal	24 Goal
Sprints	32.9 ± 2.0	30.3 ± 2.0	34.2 ± 1.2	36.4 ± 0.9	39.9 ± 2.5
Accelerations	55.6 ± 4.7	52.6 ± 4.2	51.5 ± 1.9	66.9 ± 2.0	57.0 ± 3.8
Decelerations	48.8 ± 3.3	48.7 ± 3.7	53.3 ± 2.0	60.5 ± 2.0	65.4 ± 5.3
Impacts	0.4 ± 0.2	0.8 ± 0.4	0.6 ± 0.2	1.2 ± 0.2	1.2 ± 0.3

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
