# Peer review of "The Spatiotemporal Characteristics of 0–24-Goal Polo"

_animals, 2019, doi:10.3390/ani9070446_

Round 1

Reviewer 1 Report

A really nice study which employs techniques common in human sport to undertake performance analysis in equestrian sport, which is much needed to build an evidence base for training and competition strategies, and to protect equine welfare.

Simple summary

Nice summary of study provided, some minor amendments suggested to increase clarity

Line 11: would be good to demonstrate handicap for players is out of 10, could you include (-2 to 10) after ‘handicap system’ in this line to denote this?

Line 13: suggest including ‘the’ before four

Line 13: ‘levels’ feels a little vague, could you be a little more specific here to support the reader less familiar with this discipline

Line 17: as you haven’t monitored ponies’ HR etc suggest amending to ‘handicap, they will need to ensure’

Line 20: suggest replacing ‘their’ with ‘pony’

Abstract

Line 25: please amend to ‘data were’ as data are plural

Line 28: it would be beneficial to define trivial, large and very large as individual interpretation could be quite subjective

Line 29: it would also be good here to indicate speed zones 4 and 5 are higher intensity to enable the abstract to stand alone

Line 32: suggest amending to ‘highlight increased cardiovascular, anaerobic and speed based physiological demands on Polo ponies as playing level increases’ to align with principles of training and suggest link across to training regimes (and what they need to prepare these ponies for) in next sentence

Keywords: I would also recommend including training or training regimes and equestrian or equine in your key words

Introduction

Outline of sport is beneficial for readers less familiar with polo and context of why / how GPS data could be beneficial within equestrian activities provided.

Line 39-44: it may be worth making reference to lack of use of GPS in equine sport due to restrictions for use in competition and also as GPS tends not to work in indoor arena and some locations can be challenging due to rural locations were some competitions takes place – at authors’ discretion

Line 55: some odd formatting around citation 14

Lines 54-57: I would advise dividing this sentence into two as it addresses two distinct constructs. Polo does have a large pitch – finish this element with narrative relating to benefits to study GPS (although one could argue eventing XC or endurance races present bigger challenges). Then address the increased use of racehorses as polo ponies. I agree this could relate to changes in the game but it would be worth addressing that there is also an increased focus globally to address wastage in racing and numerous schemes to rehome/ retrain racehorses for a life after racing in other equestrian disciplines, of which polo could be one.

Line 62: replace ‘research’ with ‘study’

Line 64: please amend ‘is’ to ‘was’ as the research has been completed

Line 65: please amend ‘will’ to ‘would’

Materials and Methods

Generally clear method – some additional information required to make method fully repeatable.

Lines 69-75: I am assuming these matches were field polo but would be good to clarify this in this paragraph

Line 74: may be useful to indicate the percentage and number of 16 and 24 goal games included here (xx%; n=xx); it would also be worthwhile clarifying if full or half chukka strategies were used by players for pony use within your method

Lines 83-86: whilst I agree a player mounted unit is a much more feasible approach, this does need to be considered later when translating results to polo ponies, as players will change ponies and the rules require enforced rest chukkas which would affect playing strategies, CV workload etc

Lines 86-89: repeated sentence, please remove

Line 93: is there a relevant citation, which could be included to support this approach?

Line 96: please amend to ‘data were’ and ‘were trimmed’

Line 97-98: please describe what notational analyses took place and by whom earlier in the method

Line 106: please amend to ‘data were’

Line 108: please amend to ‘data were’

Line 112: please amend ‘is’ to ‘if’ and provide CI in full here with (CI) after

It would be worthwhile to include some statistical analysis to investigate the differences reported in each of your game groups; you could undertake an omnibus test of difference with post hoc analyses to compare average speed and distance covered per chukka in each level of game relatively easily to identify if differences occur between 0, 6, 10, 16 and 24 goal games, which would complement the data reported. This would be worthwhile to undertake and add into your paper.

Results

Lines 114-116: this para feels a little wieldy; I would advise presenting the key descriptive statistics and remove the preamble, but please insert units for medians presented. assuming data were no –parametric and this is why medians are reported, but it would be useful to state this for your readers.

Line 119: suggest amending to ‘Figure 1; a predominant increase in median distance covered per chukka was seen as cumulative player handicap increased’

Line 121: please present in past tense as your study has been completed, amend ‘are’ to ‘were’ and ‘is’ to ‘was’

Line 122: suggest making a new sentence and changing to ‘On average, 10 goal chukka distances recorded a small increase in mean distance compared to the distances covered per 122 chukka at 0 and 6 goal levels’.

Line 125: suggest removing ‘above’

Line 127: please amend to past tense, ‘increased, there was a trend’

Line 128: amend ‘is’ to ‘was’ and ‘displays’ to displayed’

Line 137: amend ‘increases ‘ to ‘increased’ 

Line 155 and 156: amend ‘increase ‘ to increased’ and ‘increases’ to  ‘increase’

Line 156: please amend ‘are’ to ‘were’ and ‘have’ to ‘had’

Line 157: please amend ‘are’ to ‘were’ and ‘is’ to ‘was’

Line 158: suggest changing to ‘ play only differed trivially’

Line 159: please amend ‘differ’ to ‘differed’

Line 160: please amend ‘is’ to ‘was’

Line 161: to ‘were’

Line 162: please change 6 to six as start of sentence

Line 163: please amend ‘are’ to ‘were’

Line 165: suggest replacing ‘show’ with ‘record’

Line 167: please amend ‘are’ to ‘were’

Line 170: please amend ‘are’ to ‘were’

Line 178: please amend ‘is’ to ‘was’

Line 179: please amend ‘increases’ to ‘increased’ and ‘is’ to ‘was’

Line 181 to 182: suggest amending to ‘The only exception occurred between 0 and 181 10 goal levels where a pony would perform a small increase in decelerations.’

Line 180 and 183: please amend ‘are’ to ‘were’

Line 183: please amend ‘increases’ to ‘increased’

Line 184: please amend ‘are’ to ‘were’

Line 196: please amend to ‘occurred’ and ‘required’

Lines 189, 190, 191 and 193: please amend ‘are’ to ‘were’

Line 193: please amend ‘is’ to ‘was’

Discussion

Lines 204-209: I agree your results offer an insight into how we could manage polo ponies tactics in competition and could be translated to inform training regimes but I think you need to draw this out more in your discussion here, particularly to show the alignment to improving welfare and preventing injury through more correct management, so it is explicit to the reader – you need to show the reader how they will / could be used to best effect here

Line 235: not all readers may be familiar with ‘a string’ I would suggest including a definition, adding ‘of ponies’ or adding an explanation as a foot note

Line 237 and 238: not just fitness and anaerobic capacity but also other principles of training such as skill acquisition and perhaps talent? may be worth considering these here too

Line 253: previous work has shown half-chukkering did not supported recovery as effectively as full chukka strategies within games and was more likely to induce pony fatigue, would be worth considering this here (Williams and Fiander, 2014)

Line 256: please amend ‘was’ to ‘were’

Line 264 -266: repeated sentence please remove

Line 279: suggest amending to ‘to complement established risk management strategies’ and I would consider if the increased application of notational analysis within the sport could also beneficial here / going forwards

It would be beneficial to include a limitations section and discuss the pros / cons of using a player mounted GPS unit here.

Conclusion

Lines 286-287: suggest amending to ‘As the level of play increases, the increased average speed and distance covered require ponies to possess the cardiovascular and anaerobic performance / fitness to match the physiological demand of the level of Polo they are playing.’

Line 292-296: suggest moving future research into discussion section

Author Response

A really nice study which employs techniques common in human sport to undertake performance analysis in equestrian sport, which is much needed to build an evidence base for training and competition strategies, and to protect equine welfare.

We thank the reviewer for their kind summary of our work. This was the intention of the study, and we are pleased that this has come across.

Simple summary

Nice summary of study provided, some minor amendments suggested to increase clarity

Line 11: would be good to demonstrate handicap for players is out of 10, could you include (-2 to 10) after ‘handicap system’ in this line to denote this? Amended as requested

Line 13: suggest including ‘the’ before four Amended as requested

Line 13: ‘levels’ feels a little vague, could you be a little more specific here to support the reader less familiar with this discipline We feel that ‘levels’ is a term used across multiple sports, equestrian or otherwise, and so not only best fits with the intention of the current paper but increases the transferability of this methodology to other equestrian pursuits.

Line 17: as you haven’t monitored ponies’ HR etc suggest amending to ‘handicap, they will need to ensure’ Amended as requested

Line 20: suggest replacing ‘their’ with ‘pony’ Amended as requested

Abstract

Line 25: please amend to ‘data were’ as data are plural Amended as requested

Line 28: it would be beneficial to define trivial, large and very large as individual interpretation could be quite subjective Whilst we acknowledge the comment regarding individual interpretation, we wish to not define these magnitudes of effect within the abstract. We feel that by doing so, the abstract would become ‘clunky’ and clarity of the findings may actually be lost. We also feel the inclusion of the reference for these descriptors would not add further clarity for the reader.  

Line 29: it would also be good here to indicate speed zones 4 and 5 are higher intensity to enable the abstract to stand alone Amended as requested – this sentence now reads ‘time spent in high intensity speed zones (zones 4 and 5)…’

Line 32: suggest amending to ‘highlight increased cardiovascular, anaerobic and speed based physiological demands on Polo ponies as playing level increases’ to align with principles of training and suggest link across to training regimes (and what they need to prepare these ponies for) in next sentence Amended as requested

Keywords: I would also recommend including training or training regimes and equestrian or equine in your key words Amended as requested

Introduction

Outline of sport is beneficial for readers less familiar with polo and context of why / how GPS data could be beneficial within equestrian activities provided.

Line 39-44: it may be worth making reference to lack of use of GPS in equine sport due to restrictions for use in competition and also as GPS tends not to work in indoor arena and some locations can be challenging due to rural locations were some competitions takes place – at authors’ discretion – having consulted the rules for racing, eventing and Polo there appears to be no restrictions for use on competition of GPS technology. We performed a thorough search of relevant literature and could not find this discussed in this context even when using broad terms such as ‘technology’ ‘aid’ ‘GPS’ or ‘assistance’. Therefore, we feel this statement may be unwarranted and is unsupported at present. Further, the GPS units used within this study have an indoor mode to allow quantification of spatiotemporal characteristics and athlete load in indoor sports. We look forward to carrying out this work in equestrian sports in the future.

Line 55: some odd formatting around citation 14 – Apologies, amended as requested

Lines 54-57: I would advise dividing this sentence into two as it addresses two distinct constructs. Polo does have a large pitch – finish this element with narrative relating to benefits to study GPS (although one could argue eventing XC or endurance races present bigger challenges). Then address the increased use of racehorses as polo ponies. I agree this could relate to changes in the game but it would be worth addressing that there is also an increased focus globally to address wastage in racing and numerous schemes to rehome/ retrain racehorses for a life after racing in other equestrian disciplines, of which polo could be one. Thank you for this recommendation, the sentences now read ‘Polo is played on the largest pitch in professional sport (275m x 145m), this permits the attainment of high speeds and distances covered by Polo ponies. Furthermore, Polo is seeing an increase in ponies transitioning from racing [14]; this may increase game speed or encourage the tactical use of fast ponies, as well as promoting horse longevity by providing a viable life after racing [14].’

Line 62: replace ‘research’ with ‘study’

Line 64: please amend ‘is’ to ‘was’ as the research has been completed

Line 65: please amend ‘will’ to ‘would’ All amended as requested

Materials and Methods

Generally clear method – some additional information required to make method fully repeatable.

Lines 69-75: I am assuming these matches were field polo but would be good to clarify this in this paragraph Amended to include the term outdoor field Polo at the reviewer’s request.

Line 74: may be useful to indicate the percentage and number of 16 and 24 goal games included here (xx%; n=xx); it would also be worthwhile clarifying if full or half chukka strategies were used by players for pony use within your method

Thank you for this suggestion, we feel it aids in the clarity of the manuscript. We have included the following amendments to reflect the contribution of each level of play to the present dataset to read as follows: ‘All games were contested under Hurlingham Polo Association rules [13] and were played over four chukkas (0, 6 and 10-goal, n = 40, 40 and 80 respectively), with the exception of 16 (n = 154) and 24-goal (n = 24) games, which were contested over six chukkas.’ We have addressed the second part of this comment below, with your comment for line 83-86.

Lines 83-86: whilst I agree a player mounted unit is a much more feasible approach, this does need to be considered later when translating results to polo ponies, as players will change ponies and the rules require enforced rest chukkas which would affect playing strategies, CV workload etc

Due to the overlap of players and ponies between levels, and the varying sample sizes for each level within the present investigation we strongly feel that providing data on a per chukka basis allows for a fairer comparison between levels. This comparison is perhaps also more easily understood when data are presented as full chukkas than if half or full chukka strategies are outlined. Such strategies are completely at the discretion of the player, and can be tactical, or as the result of outside influence e.g. Pony injury and as such do not constitute a variable that can be controlled for experimentally. Whilst we agree that higher goal Polo is more likely to employ rotational horse management strategies, this was also apparent in low goal Polo but again was at the discretion of the player.

Lines 86-89: repeated sentence, please remove Amended as requested

Line 93: is there a relevant citation, which could be included to support this approach? No supporting reference is available so the following amendment has been made: ‘The belt pouch was secured with insulation tape to minimize potential oscillation of the unit during data collection’

Line 96: please amend to ‘data were’ and ‘were trimmed’ Amended as requested

Line 97-98: please describe what notational analyses took place and by whom earlier in the method Included as requested, at revised Lines 120-121, reading as follows: ‘Notational analysis was performed by the researchers during the game to describe start and end time of each chukka, which was used to ‘trim’ data subsequently.’

Line 106: please amend to ‘data were’

Line 108: please amend to ‘data were’

Line 112: please amend ‘is’ to ‘if’ and provide CI in full here with (CI) after Amended as requested, except for C.I. as this abbreviation has been introduced in full two sentences prior at revised line 138.

It would be worthwhile to include some statistical analysis to investigate the differences reported in each of your game groups; you could undertake an omnibus test of difference with post hoc analyses to compare average speed and distance covered per chukka in each level of game relatively easily to identify if differences occur between 0, 6, 10, 16 and 24 goal games, which would complement the data reported. This would be worthwhile to undertake and add into your paper.

We accept that the use of omnibus testing to obtain statistical significance is commonplace in papers similar to this one. However, as players and ponies within this sample may have played across multiple handicap levels the assumptions of parametric omnibus testing are violated in as much as the samples are not independent and not normally distributed so it is our understanding that such a test cannot be performed. Furthermore, we feel differences are best expressed as magnitude of effects and confidence around these effects as to state that differences are significant or not presents a potentially false dichotomy between cumulative handicaps – a point that is complicated by the composition of this cumulative handicap not being known.

Results

Lines 114-116: this para feels a little wieldy; I would advise presenting the key descriptive statistics and remove the preamble, but please insert units for medians presented. assuming data were no –parametric and this is why medians are reported, but it would be useful to state this for your readers. Preamble removed and units added as requested. The following sentence is included at the end of the methods section to describe the use of medians ‘Descriptive data were reported as medians, unless otherwise stated, due to the data being non-normality distributed.’

Line 119: suggest amending to ‘Figure 1; a predominant increase in median distance covered per chukka was seen as cumulative player handicap increased’ Amended as requested.

Line 121: please present in past tense as your study has been completed, amend ‘are’ to ‘were’ and ‘is’ to ‘was’

Line 122: suggest making a new sentence and changing to ‘On average, 10 goal chukka distances recorded a small increase in mean distance compared to the distances covered per 122 chukka at 0 and 6 goal levels’.

Line 125: suggest removing ‘above’

Line 127: please amend to past tense, ‘increased, there was a trend’

Line 128: amend ‘is’ to ‘was’ and ‘displays’ to displayed’

Line 137: amend ‘increases ‘ to ‘increased’

Line 155 and 156: amend ‘increase ‘ to increased’ and ‘increases’ to  ‘increase’

Line 156: please amend ‘are’ to ‘were’ and ‘have’ to ‘had’

Line 157: please amend ‘are’ to ‘were’ and ‘is’ to ‘was’

Line 158: suggest changing to ‘ play only differed trivially’

Line 159: please amend ‘differ’ to ‘differed’

Line 160: please amend ‘is’ to ‘was’

Line 161: to ‘were’

Line 162: please change 6 to six as start of sentence

Line 163: please amend ‘are’ to ‘were’

Line 165: suggest replacing ‘show’ with ‘record’

Line 167: please amend ‘are’ to ‘were’

Line 170: please amend ‘are’ to ‘were’

Line 178: please amend ‘is’ to ‘was’

Line 179: please amend ‘increases’ to ‘increased’ and ‘is’ to ‘was’

Line 181 to 182: suggest amending to ‘The only exception occurred between 0 and 181 10 goal levels where a pony would perform a small increase in decelerations.’

Line 180 and 183: please amend ‘are’ to ‘were’

Line 183: please amend ‘increases’ to ‘increased’

Line 184: please amend ‘are’ to ‘were’

Line 196: please amend to ‘occurred’ and ‘required’

Lines 189, 190, 191 and 193: please amend ‘are’ to ‘were’

Line 193: please amend ‘is’ to ‘was’

All grammatical changes amended as requested, with recommended sentence changes also incorporated.

Discussion

Lines 204-209: I agree your results offer an insight into how we could manage polo ponies tactics in competition and could be translated to inform training regimes but I think you need to draw this out more in your discussion here, particularly to show the alignment to improving welfare and preventing injury through more correct management, so it is explicit to the reader – you need to show the reader how they will / could be used to best effect here

Thank you for this comment, we agree there is a need to acknowledge the translational value of this work within the discussion/conclusion. However, if we were to do so at this point in the discussion as suggested, we feel it would be premature. Throughout the discussion and conclusion, we feel we address this comment in a more appropriate manner, allowing for consideration of nuance and providing supporting examples where appropriate. We have also removed the repeated sentence.

Line 235: not all readers may be familiar with ‘a string’ I would suggest including a definition, adding ‘of ponies’ or adding an explanation as a foot note – Amended as requested – the term ‘of ponies’ is included in the first instance to inform the reader.

Line 237 and 238: not just fitness and anaerobic capacity but also other principles of training such as skill acquisition and perhaps talent? may be worth considering these here too – we respectfully decline to include comment on skill acquisition and talent in this instance. Whilst we acknowledge that the skill-based training regime a Polo pony is exposed to is vital for their development we did not measure this, and therefore cannot comment on this occasion.

Line 253: previous work has shown half-chukkering did not supported recovery as effectively as full chukka strategies within games and was more likely to induce pony fatigue, would be worth considering this here (Williams and Fiander, 2014)

We acknowledge that the cited paper adds value to the literature, but is only somewhat relevant to the present study because each investigation measures stand alone expressions of work load undertaken. The reference provided measures internal workload, whereas we have measured external workload. Until an investigation is undertaken that assesses these parameters in unison we cannot be confident that lower measures of internal workload are not a product of lower external workload, which may partially explain the lower heart rates observed by full chukka ponies. We would be really interested in collaborating on such a paper in the future.

Line 256: please amend ‘was’ to ‘were’

Line 264 -266: repeated sentence please remove Amended as requested.

Line 279: suggest amending to ‘to complement established risk management strategies’ and I would consider if the increased application of notational analysis within the sport could also beneficial here / going forwards – Amended and included as requested.

It would be beneficial to include a limitations section and discuss the pros / cons of using a player mounted GPS unit here. Thank you for this recommendation, we have provided a short limitation/delimitation statement here, which links to future recommendations as suggested below. This section reads as follows: ‘A possible limitation of this study is the use of a player worn GPS unit which indirectly but reliably measures the characteristics outlined in this paper [7]. However, this is considered most feasible for Polo as a player potentially undertakes many pony-player interactions per game [7]. The use of a player worn GPS unit may also permit an investigation into the unique movement signature brought about by individual pony-player interactions, allowing for a thorough kinematic evaluation of riding technique and resultant pony gait. This paper has identified trends and values at a team level, however future research may seek to investigate how these metrics vary at an individual level to identify the strengths and weaknesses within a player’s string, and how best to train or manage these ponies. Further work is also required to understand whether player position interacts with measures of equine Polo performance in a causative manner.’

Conclusion

Lines 286-287: suggest amending to ‘As the level of play increases, the increased average speed and distance covered require ponies to possess the cardiovascular and anaerobic performance / fitness to match the physiological demand of the level of Polo they are playing.’ Amended as requested.

Line 292-296: suggest moving future research into discussion section Amended as above.

Reviewer 2 Report

The paper aims to assess the spatiotemporal demands of Polo and to describe the performance requirements placed upon polo ponies across varying levels of polo play.  Key findings of this investigation were that as cumulative player handicap increased, so too did distance covered per chukka, with a greater proportion of time spent at higher velocities and a greater number of high intensity activities also performed. With the increases in average speeds and distances covered as level of play increases, the cardiovascular and anaerobic needs of Polo ponies must match the demands of the level of Polo they are playing.

This paper shows the trends for increase in distance per chukka in relation to the player handicap. It would be interesting to know if there were any statistical differences between speed zones and level of play?  Only trends are noted but no statistical differences are presented.

Line 23 remove also investigated

Line 86-88 repeated sentence

Line 125 remove above

Line 158-159  remove repeat (with 10 and 24 goals also differing trivially)

Line 207-209 repeated sentence

Line 264-266 repeated sentence

Author Response

The paper aims to assess the spatiotemporal demands of Polo and to describe the performance requirements placed upon polo ponies across varying levels of polo play.  Key findings of this investigation were that as cumulative player handicap increased, so too did distance covered per chukka, with a greater proportion of time spent at higher velocities and a greater number of high intensity activities also performed. With the increases in average speeds and distances covered as level of play increases, the cardiovascular and anaerobic needs of Polo ponies must match the demands of the level of Polo they are playing.

We thank the reviewer for their kind summary of our work. This was the intention of the study, and we are pleased that this has come across.

This paper shows the trends for increase in distance per chukka in relation to the player handicap. It would be interesting to know if there were any statistical differences between speed zones and level of play?  Only trends are noted but no statistical differences are presented.

We feel that differences are best expressed as magnitude of effects and confidence around these effects in this instance. Stating whether differences are significant or not presents a potentially false dichotomy between cumulative handicaps – a point that is complicated by the composition of this cumulative handicap not being known, and the dataset containing multiple entries by the same players and ponies across different cumulative handicaps.

Line 23 remove also investigated

We decline this recommendation as these investigations were performed by separate research groups

Line 86-88 repeated sentence

Line 125 remove above

Line 158-159  remove repeat (with 10 and 24 goals also differing trivially)

This is not a repetition but a different comparison, so the sentence has been amended to read as follows: ‘In Zone 2, 0 goal play only differed trivially to that of 10 and 24 goals, with 10 and 24 goals also differing trivially to each other’

Line 207-209 repeated sentence

Line 264-266 repeated sentence

All amended as requested.

Round 2

Reviewer 1 Report

Thank you for the clear outline of the amendments you have made to the manuscript. There is one minor area I would suggest amending (below) but I feel the flow of the paper has improved and I happy the reasons articulated for not making all suggested changes are valid, and am pleased to  recommend your work for publication.

Line 119-120: suggest amending to 'being non-parametric'